# Validity and Reliability of a Wearable Goniometer Sensor Controlled by a Mobile Application for Measuring Knee Flexion/Extension Angle during the Gait Cycle

**DOI:** 10.3390/s23063266

**Published:** 2023-03-20

**Authors:** Tomoya Ishida, Mina Samukawa

**Affiliations:** Faculty of Health Sciences, Hokkaido University, North 12, West 5, Kita-ku, Sapporo 060-0812, Japan

**Keywords:** gait analysis, knee kinematics, motion analysis, wearable sensor, validation

## Abstract

Knee kinematics during gait is an important assessment tool in health-promotion and clinical fields. This study aimed to determine the validity and reliability of a wearable goniometer sensor for measuring knee flexion angles throughout the gait cycle. Twenty-two and seventeen participants were enrolled in the validation and reliability study, respectively. The knee flexion angle during gait was assessed using a wearable goniometer sensor and a standard optical motion analysis system. The coefficient of multiple correlation (CMC) between the two measurement systems was 0.992 ± 0.008. Absolute error (AE) was 3.3 ± 1.5° (range: 1.3–6.2°) for the entire gait cycle. An acceptable AE (<5°) was observed during 0–65% and 87–100% of the gait cycle. Discrete analysis revealed a significant correlation between the two systems (R = 0.608–0.904, *p* ≤ 0.001). The CMC between the two measurement days with a 1-week interval was 0.988 ± 0.024, and the AE was 2.5 ± 1.2° (range: 1.1–4.5°). A good-to-acceptable AE (<5°) was observed throughout the gait cycle. These results indicate that the wearable goniometer sensor is useful for assessing knee flexion angle during the stance phase of the gait cycle.

## 1. Introduction

Assessment of knee kinematics during gait is useful in various fields, including clinical and health promotion. Patients with knee osteoarthritis demonstrate smaller knee flexion excursion during the loading response phase and peak knee flexion, but a greater knee flexion angle at initial contact; this is considered to have long-term adverse effects on knee joint health [1]. Additionally, it has been confirmed that the knee flexion angle of patients following anterior cruciate ligament reconstruction is greater within the first six postoperative months than that in healthy controls, but smaller after 1–3 years and/or more than 3 years [2]. Researchers speculated that these altered knee kinematics during gait are associated with post-traumatic osteoarthritis following anterior cruciate ligament reconstruction [3,4]. Therefore, knee kinematics during gait has received greater attention in rehabilitation for knee disorders [5,6,7,8,9]. Recent studies of more than 100 community-dwelling older individuals have indicated that those with locomotive syndrome exhibit a smaller knee flexion angle during the swing phase of gait, which may be associated with locomotive disability [10,11]. In another study of community-dwelling older individuals, greater variability of knee kinematics during gait was found in elderly women with frailty than in those without frailty [12]. Thus, assessing knee kinematics during gait is important for detecting functional deficits in the field of health promotion [10,11,12].

Until recently, the gold standard method for assessing knee kinematics was three-dimensional motion analysis using optical tracking systems, which are widely used in laboratory settings. However, these systems have several limitations including the requirement for a specific recording space, significant time investment for data acquisition and analysis, and a high cost. Additionally, long-time monitoring of kinematics during daily living is not available. Therefore, the availability of optical motion analysis systems in clinical and health promotion fields is very limited.

A wearable goniometer is a system for assessing the knee flexion angle during gait [13,14]. Furthermore, wearable goniometers can be controlled, even by mobile applications [15], as angle calculations using an electro-goniometer are much simpler than those using optical tracking. Thus, wearable goniometers have the potential to be widely used in clinical and health promotion fields. However, accuracy is a concern because the goniometer can be influenced by the attachment location and/or lower limb alignment, compared with other wearable systems, such as inertial measurement unit (IMU)-based systems [14,16]; issues are more pronounced at a flexed position [15,16]. The knee joint is not an ideal revolute joint, as axial rotation and anterior-posterior translation occur with knee extension/flexion motion, and the centre of rotation at the knee joint changes with knee extension/flexion angle [17]. Therefore, measurements recorded during the stance phase of gait, when the knee flexion angle is relatively small, may be more accurate than those recorded during the swing phase, when the knee flexion angle is large. However, previous studies have reported the validity of limited discrete values, such as peak flexion or extension angle, and mean values for a gait cycle [13,15,18,19]. However, it remains unclear how the validity and reliability of the wearable goniometer change throughout the gait cycle. Assessment of the entire kinematic trajectory has focused on detecting patient characteristics [20,21].

Consequently, the primary purpose of the present study was to determine the validity of the measurements recorded by a wearable goniometer for the assessment of knee flexion angle throughout the gait cycle, in addition to discrete value analysis, compared with a three-dimensional motion analysis system based on optical tracking as the gold standard. The secondary aim was to determine the intra-rater reliability of a wearable goniometer in assessing the knee flexion angle during the gait cycle. We hypothesised that greater validity and reliability would be observed during the stance phase than during the swing phase of the gait cycle.

## 2. Materials and Methods

### 2.1. Study Design

This study included two components, a validation study and a reliability study. In the validation study, knee kinematics during gait were simultaneously assessed using a wearable goniometer and three-dimensional motion analysis, based on an optical tracking system. The two measurement methods were compared throughout a gait cycle using the gold standard, three-dimensional motion analysis, as a reference. The test–retest reliability of a wearable goniometer for a single tester with a 1-week interval between the two measurement days (Day 1 and Day 2) was assessed in the reliability study. The two studies were conducted separately.

### 2.2. Participants

In total, 21 participants (10 male and 11 female participants; age: 22.3 ± 1.9 years; body height: 163.7 ± 8.8 cm; body weight: 56.7 ± 8.2 kg) were enrolled in the validation study, and 17 participants (7 male and 10 female participants; age: 22.1 ± 1.5 years; body height: 161.7 ± 7.8 cm; body weight: 55.5 ± 6.0 kg) were enrolled in the reliability study. The exclusion criteria were knee pain during daily activity, history of lower-extremity surgery, and other disorders that could interfere with data acquisition. Written informed consent was obtained from the participants, and this study was approved by the Institutional Review Board of the Faculty of Health Sciences, Hokkaido University (approval number: 21-74).

### 2.3. Assessment Using a Wearable Goniometer

A wearable goniometer sensor (Smart Knee; Nitto Denko Corporation, Osaka, Japan), evaluating single-axis motion using a flexible bending sensor, was attached to the lateral side of the left knee of the participants using double-sided tape and elastic tape (Figure 1). The midpoint of the goniometer was adjusted to the lateral femoral epicondyle, referred to as the flexion/extension axis [22]. Data were acquired using a customised mobile application (SK App, Nitto Denko Corporation) via Bluetooth connection. The goniometer has a repeatability of less than 1 degree, and the output has a resolution of 0.1 degree. The sampling rate can be adjusted between 1 and 200 Hz.

Calibration trials were performed in standing and sitting positions. The knee flexion angle measured during the standing trials was defined as neutral (i.e., zero degrees). The knee flexion angle in the sitting position was measured via digital photography using the mobile application (Figure 2). The knee flexion angle during the motion task was determined using the linear fitting coefficient obtained from the two calibration trials on the mobile application. The knee flexion angle was sampled at 200 Hz, exported as a database file, and then as a CSV file with sensor address, time stamp, and trial name. The data were smoothed using a fourth-order Butterworth lowpass filter with a cut-off frequency of 6 Hz at post-processing using MATLAB (MathWorks, Natick, MA, USA).

### 2.4. Assessment Using a Standard Optical Motion Analysis System

Based on the process described in a previous study [23], retroreflective markers were placed on the iliac crest, anterior and posterior superior iliac spines, medial and lateral femoral epicondyles, medial and lateral malleoli, second metatarsal head and base, fifth metatarsal head, and heel. Marker clusters were also attached to the lateral part of the thigh and shank under the wearable goniometer (Figure 1). After recording the static standing trial to build each participant’s model, the medial and lateral femoral epicondyle markers were removed. The wearable goniometer was attached to the base of the thigh and shank marker clusters, which were used for tracking in gait trials. A three-dimensional motion analysis system (Cortex version 5.0.1, Motion Analysis Corporation, Santa Rosa, CA, USA) with seven cameras (Hawk cameras, Motion Analysis Corporation) was used to record the marker trajectories. The sampling rate was set to 200 Hz. Kinematic analysis was performed using Visual3D (version 6; C-Motion, Inc., Germantown, MD, USA) (Figure 3). The marker trajectories were low-pass filtered using a fourth-order Butterworth filter with a cut-off frequency of 6 Hz [24]. The thigh coordinate system was defined using the hip joint centre [25] and medial/lateral epicondyle markers. The shank coordinate system was defined using medial/lateral epicondyle and medial/lateral malleoli markers [26]. The *X*-axis (mediolateral) of the thigh was oriented from the medial epicondyle to the lateral epicondyle, and the *Z*-axis (inferior-superior) was oriented from the midpoint of the medial/lateral epicondyle to the hip joint centre. The *Y*-axis (anteroposterior) was oriented forwards, perpendicular to the *X*- and *Y*-axes. The *X*-axis (mediolateral) of the shank was oriented from the medial malleoli to the lateral malleoli, the *Z*-axis (inferior-superior) was oriented from the midpoint of the medial/lateral malleoli to the midpoint of the medial/lateral epicondyles, and the *Y*-axis (antero-posterior) was oriented forward, perpendicular to the *X*- and *Y*-axes. The knee flexion angle was calculated using a joint coordinate system with a Cardan *X-Y-Z* sequence (flexion/extension was the first) [27,28]. The angle measured during the static standing trial in each participant, defined as neutral (i.e., zero degrees), was used as the baseline for comparison with the wearable goniometer measurements.

### 2.5. Data Collection

The retroreflective markers were attached to the participants of the validity study after a five minute warm-up on a bicycle ergometer [23] to prevent the possibility of the attachment position of the pelvic markers on clothing shifting during the warm-up. The wearable goniometer was attached once the static standing trial for the optical motion analysis system was completed, and then two calibration trials for the wearable goniometer were recorded in the standing and sitting positions. Gait trials were performed on a 6 m walkway for the validity study and a 10 m walkway for the reliability study, while the gait speed was adjusted according to the preference of each participant. Three trials were recorded for the validity study and one trial was recorded for the reliability study—after two to three practice trials. In the reliability study, the retest was conducted by the same tester approximately 1 week after the initial data collection. The optical motion analysis system and wearable goniometer were not mechanically synchronised.

### 2.6. Data Analysis

Data processing and reduction were performed using MATLAB software. Gait data were normalised to 101 data points for the gait cycle (i.e., 0–100%), defined as the period from peak knee extension during the swing phase to that during the next swing phase. This time point was designated as the initial contact (IC) since this study aimed to investigate the validity and reliability of the wearable goniometer for measuring knee flexion angle during the gait cycle. The following knee flexion/extension angles were then derived: (1) at IC, (2) peak knee flexion during the first half of the stance phase, (3) peak knee extension during the second half of the stance phase, and (4) peak knee flexion during the swing phase (Figure 4). These discrete values are frequently used for gait assessment [1,2,11,15]. All data were averaged from the gait cycle of the three trials for the validity study, and from the mid-three gait cycles for the reliability study.

### 2.7. Statistical Analysis

The waveforms of the knee flexion angle during a gait cycle were compared between the optical motion analysis system and wearable goniometer using the coefficient of multiple correlation (CMC) for comparison of different systems [29] and the absolute error (AE) for each gait cycle. The CMC was interpreted as follows [30]:CMC ≥ 0.95: excellent0.95 > CMC ≥ 0.85: very good0.85 > CMC ≥ 0.75: good0.75 > CMC ≥ 0.65: moderate

The AE was interpreted as follows [31,32]:AE ≤ 2°: good accuracy2 < AE ≤ 5°: acceptable accuracy5° < AE ≤ 10°: tolerable accuracyAE > 10°: unacceptable accuracy

The peak knee flexion/extension angle was also compared using Pearson’s correlation coefficient. Pearson’s correlation coefficient was interpreted as follows [33]:*R* ≥ 0.9: extremely large0.7 ≤ *R* < 0.9: very large0.5 ≤ *R* < 0.7: large0.3 ≤ *R* < 0.5: moderate0.1 ≤ *R* < 0.3: small*R* < 0.1: trivial

Intra-rater reliability was assessed using the CMC for between-day assessments and AE [34], and that of the peak knee flexion/extension angle was assessed using the intraclass correlation coefficient (ICC) (1, k). The ICC was interpreted as follows [35]:ICC ≥ 0.75: excellent0.4 ≤ ICC < 0.75: fair to goodICC < 0.4: poor

Statistical analyses were performed using IBM SPSS Statistics 22 (IBM, Armonk, NY, USA). The statistical significance level was set at *p* < 0.05.

## 3. Results

### 3.1. The Validation Study

The similarity of the waveform of the knee flexion angle during the gait was excellent between the optical motion analysis and the wearable goniometer (CMC = 0.988 ± 0.012, range: 0.952–0.998) (Figure 5a). The average AE during the entire gait cycle was 3.3 ± 1.5° (range: 1.3–6.2°). Acceptable accuracy (AE < 5°) was observed during 0–65% and 87–100% of the gait cycle. The 0–6%, 25–40%, and 100% of the gait cycle were interpreted as having good accuracy (AE < 2°), and the AE during 66–86% of the gait cycle was interpreted as tolerable accuracy (AE ≤ 10°) (Figure 5b). Discrete value analysis revealed a significant correlation between the two measurement systems (*R* = 0.608–0.904, *p* ≤ 0.003) (Figure 6). A very large coefficient was noted for the knee flexion angle at IC, peak knee flexion angle during the stance phase, and peak knee extension angle during the stance phase (*R* = 0.819–0.904, *p* < 0.001) (Figure 6a–c). Additionally, a large coefficient was noted for the peak knee flexion angle during the swing phase (*R* = 0.608, *p* = 0.003) (Figure 6d).

### 3.2. The Reliability Study

The CMC between the waveforms of Day 1 and Day 2 (one week later) was 0.988 ± 0.024, and all participants, except for one, displayed excellent similarity (range: 0.898–0.999) (Figure 7a). The average AE during the entire gait cycle was 2.5 ± 1.2° (range: 1.1–4.5°). Acceptable accuracy was observed through most of the gait cycle phases, except for the 0–13%, 26–51%, and 93–100% of the gait cycle, which exhibited good accuracy (Figure 7b). Discrete value analysis showed good to excellent reliability (Figure 8). Excellent reliability was observed for the knee flexion angle at IC, peak knee flexion angle during the stance phase, and peak knee extension angle during the stance phase (ICC (1,3) = 0.880–0.927, *p* < 0.001) (Figure 8a–c). Good reliability was observed regarding the peak knee flexion angle during the swing phase (ICC (1,3) = 0.713, *p* = 0.007) (Figure 8d).

## 4. Discussion

The findings of this study suggest that the wearable goniometer used in this study is a valid and reliable tool for measuring the knee flexion angle during gait, particularly during the stance phase. Additionally, they support our a priori hypothesis and provide fundamental information regarding the usage of wearable goniometers.

The CMC between the two waveforms obtained from each measurement method was excellent for all participants. The AE of the wearable goniometer relative to the optical motion analysis system was 3.3° on average over the entire gait cycle. Previous studies have reported that the mean AE during a gait cycle was 2.3–3.3° between a goniometer system and an optical motion analysis system [13] and that the mean root-mean-square-error (RMSE) during a gait cycle was 7.1–9.3° between an IMU-based system and an optical motion analysis system [36,37], which is comparable to that reported by the present results. A discrete value analysis of the peak knee extension/flexion angle during the gait cycle also showed very large correlation coefficients between the wearable goniometer and optical motion analysis system for knee flexion angle at IC, and peak knee flexion and extension during the stance phase. However, the present study found that the AE was smaller during the stance phase than during the swing phase. AE was acceptable (AE ≤ 5°) through 0–65% and 87–100% of the gait cycle, while AE was tolerable (5° < AE ≤ 10°) during 66–86% of the gait cycle (corresponding to the swing phase). Thus, even if the mean AEs for a gait cycle are acceptable, caution is required in interpreting the results because the error may be large in certain phases (e.g., the swing phase in this study). A previous study also showed that RMSE for maximum knee flexion angle during a gait cycle was 9.1°, between an IMU-based system and an optical motion analysis system [32]. These findings indicate that the wearable goniometer used in this study has acceptable validity for assessing the knee flexion angle during the stance phase; however, the knee flexion angle during the swing phase and the mean error over a gait cycle should be interpreted with caution.

The reliability study also exhibited excellent CMC between Day 1 and Day 2 (1 week later) of the wearable goniometer testing, except for one participant. These results reflect previous three-dimensional gait analysis studies based on optical tracking and inertial sensor systems [31,36,37]. The AE between the two tests was interpreted as good (AE ≤ 2°) to acceptable (2 < AE ≤ 5°) throughout the gait cycle [31,32], which was comparable to a previously reported RMSE of 6.3°and 5.0° for an IMU-based and an optical motion analysis system, respectively [36]. Regarding discrete value analysis, ICCs for the knee flexion angle at IC and peak knee flexion and extension during the stance phase were excellent. Thus, the assessment of the knee flexion angle using a wearable goniometer is highly reliable.

As previously mentioned, the AE between the wearable goniometer and the optical motion analysis system was tolerable during 66–86% of the gait cycle (corresponding to the swing phase). A previous study also reported unacceptable accuracy (AE > 10°) for the excursion from the peak knee flexion to the peak knee extension during the swing phase [15]. The correlation coefficient for the peak knee flexion angle during the swing phase was interpreted as large, but was smaller than that during the stance phase. Considering the results of both studies, although some trends in the differences between the participants were captured, the measurements recorded by the wearable goniometer during the swing phase should be interpreted with caution [31,32]. A factor possibly contributing to the discrepancies of the AE and correlation coefficient between the stance and swing phases could be the magnitude of the knee flexion angle of the two phases. At a large knee flexion angle, kinematic crosstalk effect was a concern when measured with an electrical goniometer [16]. Additionally, the calibration angle in the flexed position may also contribute to the discrepancies. In the present study, the knee flexion angle was calculated using a linear fitting model determined by standing and sitting calibration trials. The knee flexion angle recorded during the sitting calibration trial might have been too small to capture the knee flexion motion during gait, since it was performed in the participant’s preferred sitting position. However, compared with the AEs recorded in a previous study on a wearable goniometer [15], the smaller AEs recorded in the present study may be due to the calibration being conducted with the sensor attached to the body. Because the centre of rotation of knee flexion-extension varies with the flexion angle [17], calibration while wearing goniometer sensors may reduce the effect of changes in the rotation centre on the errors. Differences in errors between the calibration methods should be verified in future studies. In addition, the optimal position for sensor placement has not been determined and warrants further study, although the centre of the sensor was aligned with the lateral femoral epicondyle in this study. Another possible factor contributing to more significant errors during the swing phase compared with the stance phase is the difference in knee flexion angular velocity. The greater angular velocity during the swing phase may have affected the behaviour of the flexible bending sensor during the swing phase. Future studies are needed to investigate the effect of gait speed on the validity and the reliability.

Knee flexion/extension motion analysis during gait is important for assessing knee disorders, such as knee osteoarthritis, anterior cruciate ligament reconstruction [1,2,3,4,5,6,7,8,9], and elderly locomotive disability [10,11,12]. However, previous studies on knee kinematics during gait were mostly conducted in laboratory settings and thus may not fully reflect the patient’s usual gait kinematics. Contrarily, the wearable goniometer used in this study may provide further information on the treatment and prevention of these disorders since it allows long-term monitoring, as it can be used in various environments throughout the day and is not influenced by the drift effect. In addition, the wearable goniometer and mobile application used in the present study could also display the knee flexion angle in real time (Figure 9). Real-time feedback on the range of motion expands the possibility for rehabilitation, not only with therapists but also with telehealth as a self-care tool. In addition, it could prove valuable, especially for patients with knee osteoarthritis or those who have undergone anterior cruciate ligament reconstruction, which alters knee extension/flexion kinematics during the stance phase [1,2]. Future research should be conducted utilising the advantages of wearable goniometers.

This study had some limitations. First, only healthy young adults were included; however, the validity and reliability of the measurement of the wearable goniometer may differ in other populations because of lower limb deformities that can affect the measurements of electrical goniometers [16]. Second, the validity and reliability of tasks other than gait remain unclear. A previous study reported that the errors were more significant during a deep-knee-bend task than during gait and sit-to-stand tasks [13]. Third, the most optimal attachment position of the goniometer was not firmly established in this study and the attachment position may influence the validity of the results. Fourth, this study was designed so participants would adopt their own pace. We did not control gait speed, which may have influenced the results. Finally, the lack of other systems to use as a benchmark is a limitation of this study.

## 5. Conclusions

The validity study showed that the CMC of the knee flexion waveform during gait between the wearable goniometer and the optical motion analysis system was 0.992 ± 0.008 (0.969–0.998). The AE between the two systems was 3.3 ± 1.5° (range: 1.3–6.2°) for the entire gait cycle, and was less than 5° for 0–65% and 87–100% of the gait cycle. The reliability study confirmed that the CMC between the two tests (Day 1 and Day 2 with a 1-week interval) was 0.988 ± 0.024 (0.898–0.999) and AE was 2.5 ± 1.2°. Good to acceptable accuracy was confirmed throughout the gait cycle (AE < 5°). These findings indicate that the wearable goniometer has acceptable validity and reliability for assessing the knee flexion angle during the stance phase, whereas the knee flexion angle during the swing phase should be interpreted with caution.

## Figures and Tables

**Figure 1 sensors-23-03266-f001:**
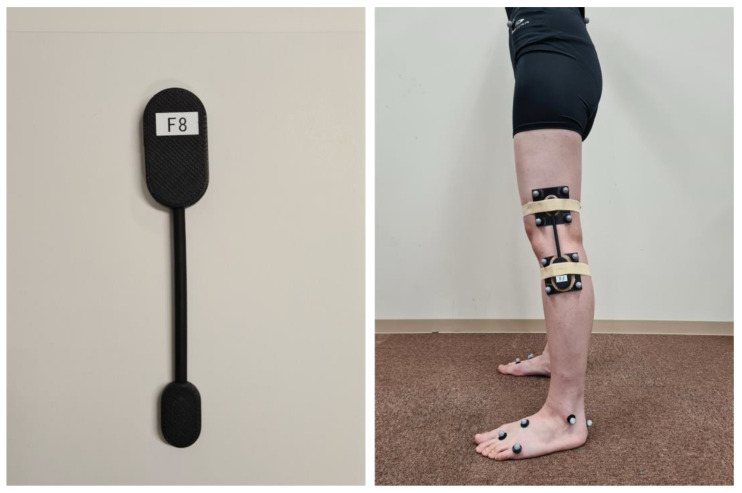
Attachment of the wearable goniometer.

**Figure 2 sensors-23-03266-f002:**
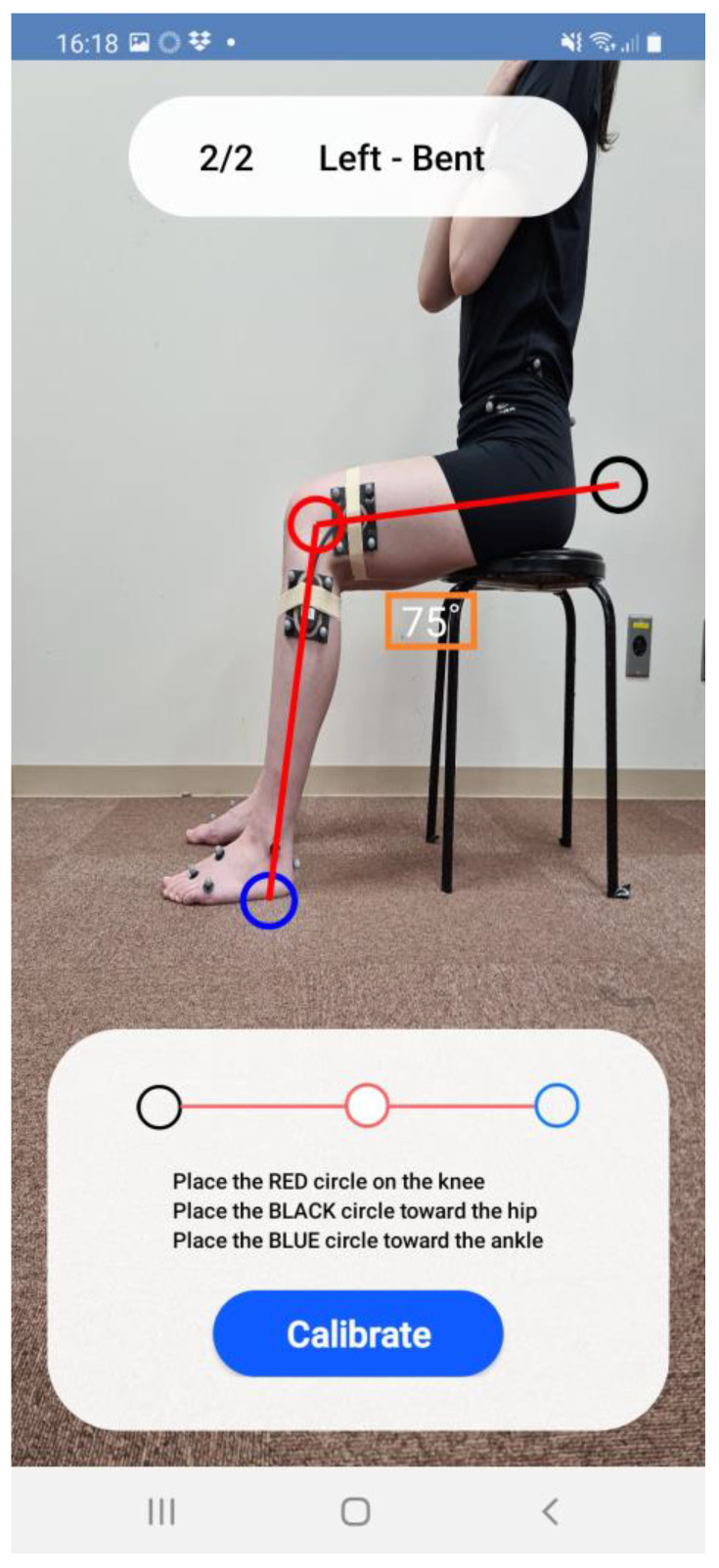
Photograph calibration of the wearable goniometer in the sitting position.

**Figure 3 sensors-23-03266-f003:**
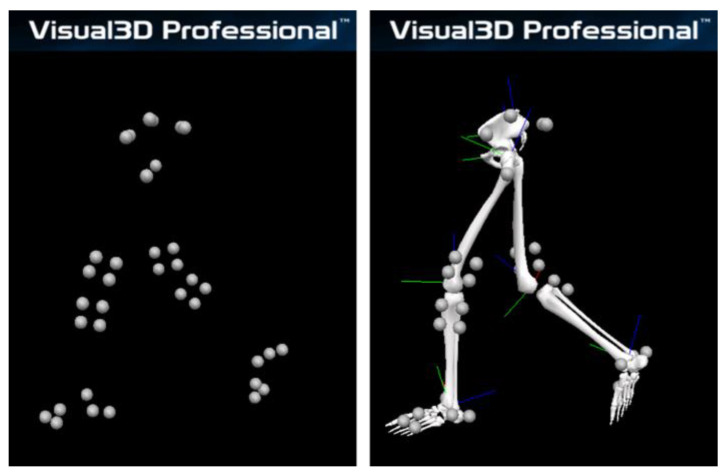
Kinematics analysis using an optical motion analysis system. Each segment coordinate system was defined from the marker coordinates, and the knee joint angle was calculated from the relative positions of the thigh and shank segments.

**Figure 4 sensors-23-03266-f004:**
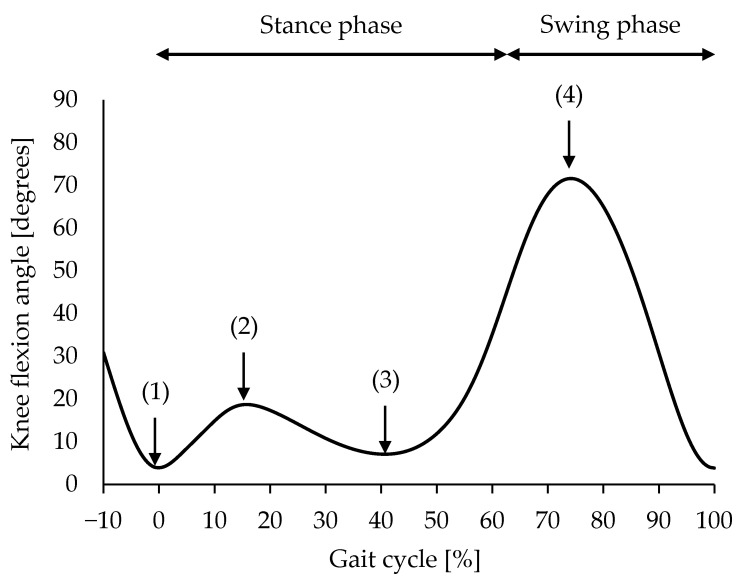
Gait cycle and discrete values of interest for analysis. The gait cycle was normalised as the period from peak knee extension during the swing phase (1) to that during the next swing phase (0–100%). For discrete value analysis, the peak knee extension angle during the swing phase (1), peak knee flexion during the first half of the stance phase (2), peak knee extension during the second half of the stance phase (3), and peak knee flexion during the swing phase (4) were derived.

**Figure 5 sensors-23-03266-f005:**
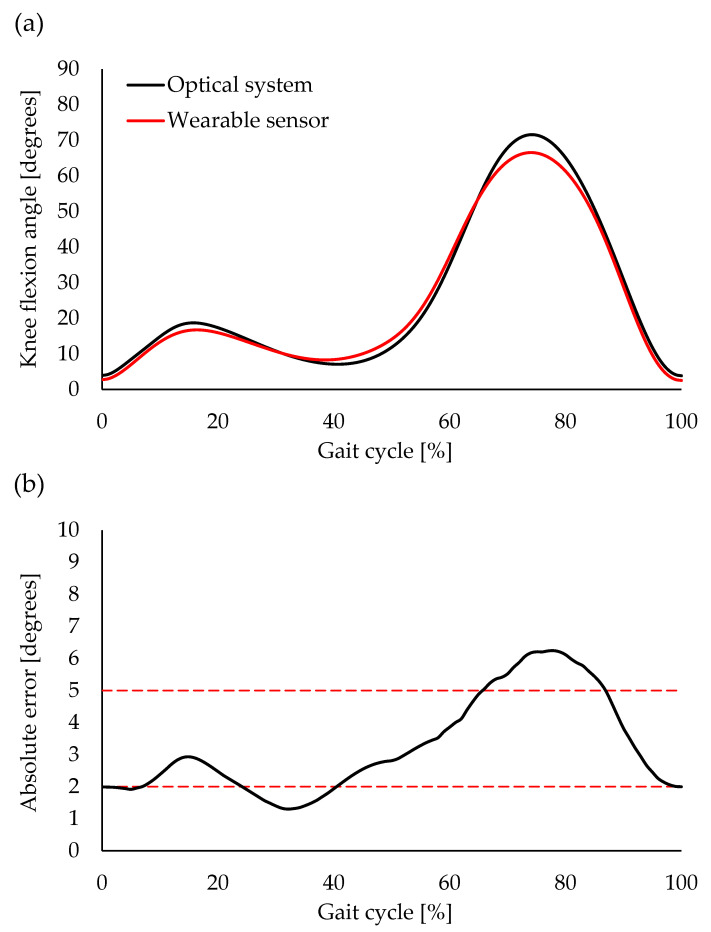
Average waveforms of knee flexion angle during a gait cycle for optical motion analysis and wearable goniometer (**a**), and absolute error (AE) between the optical motion analysis and wearable goniometer (**b**). Broken red lines indicate thresholds of interpretation for AE (AE ≤ 2° was good accuracy, 2 < AE ≤ 5° was acceptable accuracy, and 5° < AE ≤ 10° was tolerable accuracy).

**Figure 6 sensors-23-03266-f006:**
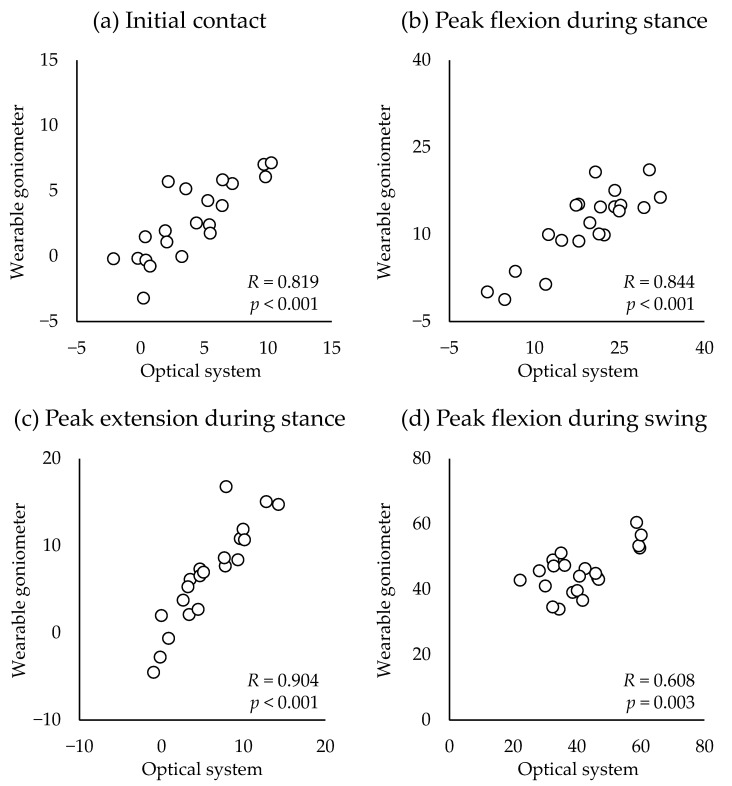
Correlations of the discrete values between optical motion analysis system and wearable goniometer.

**Figure 7 sensors-23-03266-f007:**
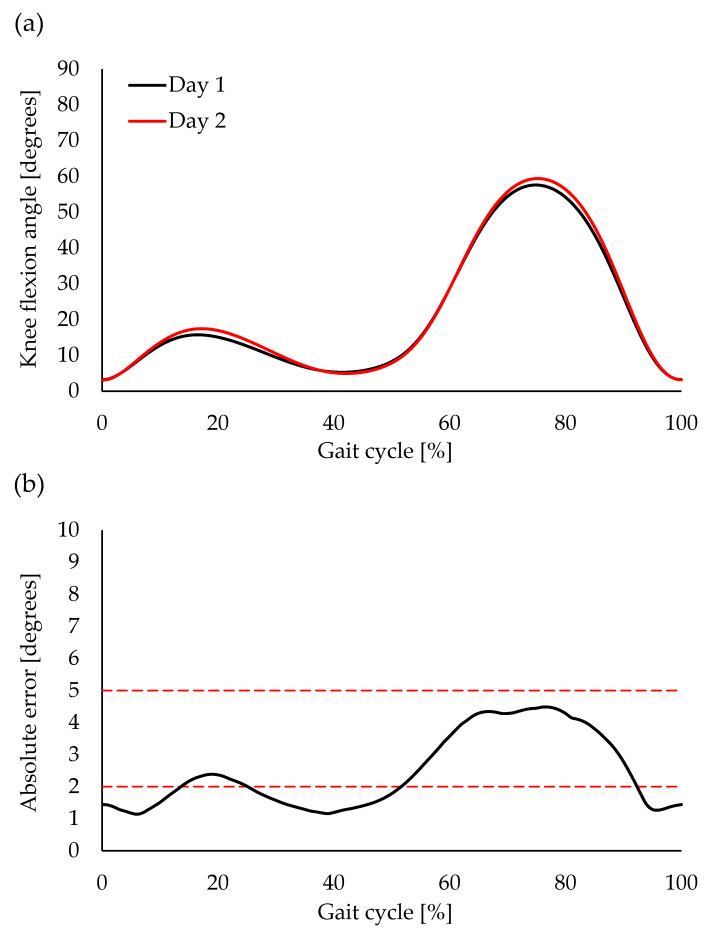
Average waveforms of knee flexion angle during a gait cycle on Day 1 and Day 2 (1 week later) of wearable goniometer testing (**a**), and absolute error (AE) between the two tests (**b**). Broken red lines indicate thresholds of interpretation for AE (AE ≤ 2° was good accuracy and 2 < AE ≤ 5° was acceptable accuracy).

**Figure 8 sensors-23-03266-f008:**
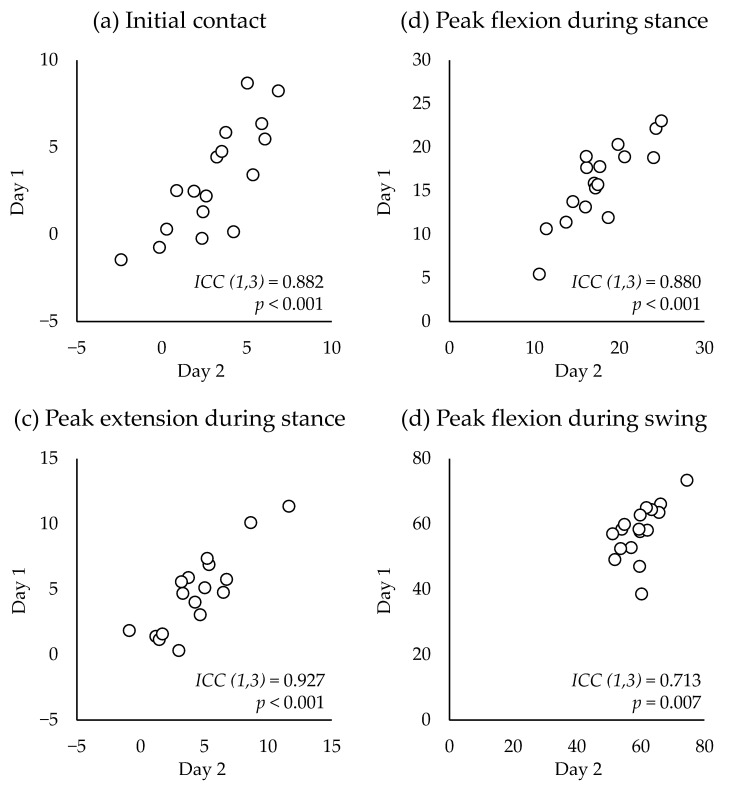
Correlations of the discrete values between Day 1 and Day 2 (1 week later) of wearable goniometer testing.

**Figure 9 sensors-23-03266-f009:**
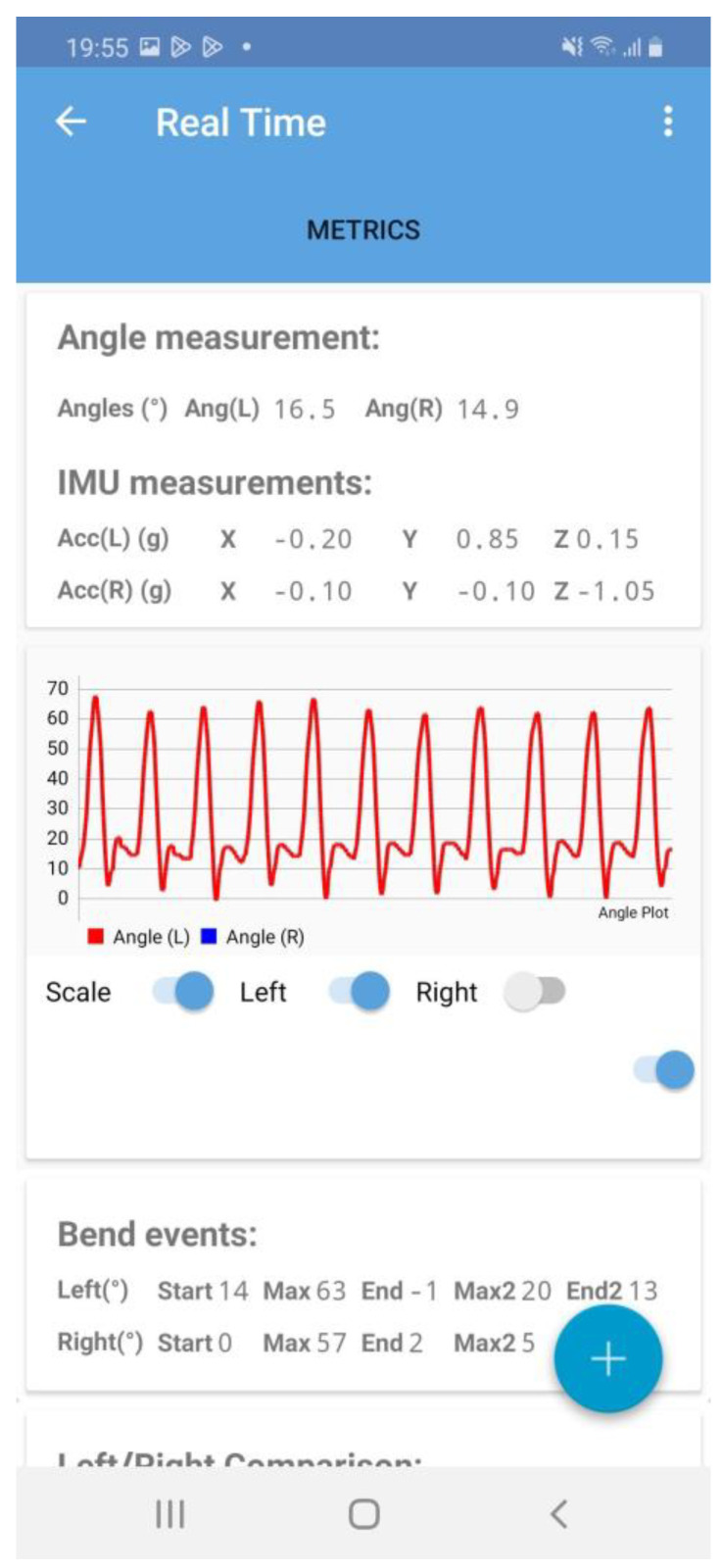
Real-time display of the knee flexion angle on the mobile application.

## Data Availability

The datasets used and/or analysed during the current study are available from the corresponding author on reasonable request.

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
