# Peer review of "Validity and Reliability of a Wearable Goniometer Sensor Controlled by a Mobile Application for Measuring Knee Flexion/Extension Angle during the Gait Cycle"

_sensors, 2023, doi:10.3390/s23063266_

Round 1

Reviewer 1 Report

Please find in the following paragraphs the review comments:

1. When the authors say: "and the AE during 6686% of the gait cycle was interpreted as 192 tolerable accuracy (AE ≤ 10°) (Figure 5b)", maybe they refer to Figure 4b instead of 5b.

2. The following paragraph is also confusing:

"A very large coefficient was noted for the knee flexion angle at IC, peak knee flexion angle 195 during the stance phase and peak knee extension angle during stance phase (R = 0.819 196 0.904, P < 0.001) (Figure 5ac)" since the: peak knee flexion angle during the stance phase is apparently shown in Fig. 5b and the peak knee extension angle during stance phase is apparently shown in Fig. 5c. However as shown above, authors mention Figure 5a-c.

3. More information about the mobile applications should be given. Is it able in a git-hub repository?

4. TYPOS:

..using mobile the mobile application..

Reviewer 2 Report

This study seems to be of rather limited value. There are numerous works on the use of goniometry in dynamic assessment of gait - particularly the extrapolation of knee angle data. 

What this study seems to be - is an assessment for a specific goniometry system that funded the project. There are numerous issues with this, beyond the declared conflict of interest statement. I will do my best to document my issues with the manuscript as presented: 

1. In the introduction, you discuss the field of health promotion and its relevance to this content. Do you have meaningful citations to back this up? 

2. IMUs can certainly have issues, like accelerometer drift, but these issues are well documented and there are numerous means of resolving them. Additionally, the author's claims that IMU systems are prohibitively expensive does not seem relevant - or at least has not been compellingly discussed. How does an IMU system compare in cost to the goniometry unit supplied to the researchers? Why couldn't a cheaper IMU-based goniometry system not be innovated (they have)? If you discuss goniometry in the introduction as you have, why not compare the performance of the goniometry system with IMUs and Motion Capture (MoCap)?

3. The authors fail to document some important issues with knee-goniometry in the introduction. That is - the knee is not an ideal revolute joint. There is no discussion of the IMPORTANT underlying biomechanics of the knee joint - which greatly limits knee joint angle measurements via goniometry. 

4. In the methodology, the authors have not adequately described sensor placement. The authors state that the midpoint of the goniometer is placed over the lateral femoral epicondyle. Why is this chosen for the center? Are there perhaps better locations to center the goniometer based on other research? Is there any citation for the selection of this locations placement and procedure for its placement? 

5. You placed the retroreflective markers on participants after a five-minute warm-up on a bicycle. Why? Cite your reasons for this procedure, or clearly justify it.

6. You have users walk at self-selected paces, why not use treadmills to better control the data produce? 

7. Figure 3 discusses the phases of gait and knee flexion angles, but the phases are discussed via points - rather than phases. This is a bit unclear and would not be intuitive for the uninitiated reader. 

8. The authors put a lot of emphasis on the CMC and the "similarity of waveform." Is this actually meaningful? A lot of measurements could do a good job of morphologically matching gait. 

9. From all of the graphics, it appears that the error greatly increases with an increase in dynamics (rate of change of knee angle). While the authors conjectured that this could be due to limitations with measuring knee angle at high flexion - there are a lot of reasons. The sensor could fail to operate well at such a rate of change, additionally the authors could probably situate the sensor in such a way that the average center of rotation is better captured. 

10. The authors purport that the wearable goniometer used for study (the one that was externally funded) is both a valid and reliable tool for measuring knee flexion angle during gate... particularly during stance. There are a number of problems with this - again some of it that could be related to CoI, where you have not adequately compared this device to any substantive benchmarking that could declare it a valid and reliable tool. Additionally, how useful is it that this device is particularly useful in the stance phase? Many low-resolution and cheap devices would be reliable in the stance phase... 

11. Again, it is misleading to use the CMC of the two waveforms as a means of identifying the goniometer's utility. There is a meaningfully high error in readings throughout the swing-phase of gait. 

Reviewer 3 Report

The manuscript presents Validity and reliability of a wearable goniometer sensor controlled by a mobile application for measuring knee flexion/extension angle during the gait cycle. This study has a simple potential engineered application. However, several concerns in the current stage of manuscript need to be addressed, as follows:

1. Authors should state wearable goniometer, optical tracking system working process or mechanism in detail. What’s more, it should provide the system photos?

2. Authors should supplement the wearable goniometer sensor data analysis model or mathematical deducing process.

3. Authors should highlight this manuscript light spots. What is the innovations, sensor fabrication, data algorithm, mobile application or just assemble the existed devices and App?

4. In the introduction, Authors should mention some wearable sensors, such as Cell Reports Physical Science 2023, 4:101191, Nano Energy 2020, 67: 104228.

Reviewer 4 Report

The paper presents the validity and the reliability of a third party wearable goniometer. The experiments are carried out following proper methodologies and adequate sample set diversity. However, the significance of the contents presented is questionable as the goniometer is a third-party one. The contribution is only in validating the data of the goniometer.

The following concerns are there that will need to be updated.

1. Details of the goniometer is not adequately presented. It is required to give key parameters such as precision, resolution, sample rate range (not limited to).

2. There are some figures that are not referred to in the main text. All figures will have to be referred to in the text.

3. In section 2.7: it is better to illustrate the data in tabular form to make them better understood.

4. There are some English mistakes found here and there in the paper.

Round 2

Reviewer 2 Report

The paper, as presented, is now of acceptable quality. I believe its contribution is marginal and I hope that the authors work to expand upon this study and benchmark this device versus comparable technology. 

Author Response

Thank you very much for your valuable comment. We will continue to study this subject and show more benefits of the goniometer sensor.

Reviewer 3 Report

I recommend the authors revise the manuscript based on my comments carefully.

Author Response

Response to reviewer 3
General Comment
I recommend the authors revise the manuscript based on my comments carefully.

Authors’ response
Thank you for reviewing our manuscript. We have modified the manuscript according to your comments. We hope the revised manuscript is now suitable for publication. The changes in the manuscript are indicated with the track changes function of Microsoft Word.

Comment 1
Authors should state wearable goniometer, optical tracking system working process or mechanism in detail. What’s more, it should provide the system photos?
Comment 2
Authors should supplement the wearable goniometer sensor data analysis model or mathematical deducing process.

Authors’ responses 1 and 2
In addition to the modifications from the first review stage, figures and descriptions have been added as follows:

Line 107: The knee flexion angle during the motion task was determined using the linear fitting coefficient obtained from the two calibration trials on the mobile application. The knee flexion angle was sampled at 200 Hz, exported as a database file and then as a CSV file with sensor address, time stamp, and trial name. The data were smoothed using a fourth-order Butterworth lowpass filter with a cut-off frequency of 6 Hz at post-processing using MATLAB (MathWorks, Natick, MA, USA).

Figure 1. Attachment of the wearable goniometer.

Line 136: The X-axis (mediolateral) of the thigh was oriented from the medial epicondyle to the lateral epicondyle, the Z-axis (inferior-superior) was oriented from the midpoint of the medial/lateral epicondyle to the hip joint centre. The Y-axis (anteroposterior) was oriented forwards, perpendicular to the X- and Y-axes. The X-axis (mediolateral) of the shank was oriented from the medial malleoli to the lateral malleoli, the Z-axis (inferior-superior) was oriented from the midpoint of medial/lateral malleoli to the midpoint of medial/lateral epicondyles, and Y-axis (antero-posterior) was oriented forward, perpendicular to the X- and Y-axes. The knee flexion angle was calculated using a joint coordinate system with a Cardan X-Y-Z sequence (flexion/extension was the first) [27,28].

Figure 3. Kinematics analysis using an optical motion analysis system. Each segment coordinate system was defined from the marker coordinates, and the knee joint angle was calculated from the relative positions of the thigh and shank segments.

Comment 3
Authors should highlight this manuscript light spots. What is the innovations, sensor fabrication, data algorithm, mobile application or just assemble the existed devices and App?

Authors’ response 3
The primary purpose of the present study was to determine the validity of wearable goniometer measurements for assessing knee flexion angle throughout the gait cycle, compared to those of a three-dimensional motion analysis system based on an optical tracking system. The main finding of this study is that wearable goniometer measurements are valid, and the tool is reliable for measuring knee flexion angle during gait, particularly during the stance phase. The wearable goniometer and the application could be used for detecting problems and therapeutic effects real time in clinical situations. In addition to the modifications in the R1 stage, we have revised some parts of the text as follows:

Line 42: Until recently, the gold standard method for assessing knee kinematics is the three-dimensional motion analysis using optical tracking systems, which is widely used in laboratory settings.

Line 67: Consequently, the primary purpose of the present study was to determine the validity of the measurements recorded by a wearable goniometer for the assessment of knee flexion angle throughout the gait cycle, in addition to discrete value analysis, compared to a three-dimensional motion analysis system based on optical tracking as the gold standard.

Line 77: In the validation study, knee kinematics during gait were simultaneously assessed using a wearable goniometer and three-dimensional motion analysis, based on an optical tracking system. The two measurement methods were compared throughout a gait cycle using the gold standard, three-dimensional motion analysis, as a reference.

Line 286: However, the present study found that the AE was smaller during the stance phase than during the swing phase. AE was acceptable (AE ≤ 5°) through 0–65% and 87–100% of the gait cycle, while AE was tolerable (5° < AE ≤ 10°) during 66–86% of the gait cycle (corresponding to the swing phase). Thus, even if the mean AEs for a gait cycle are acceptable, caution is required in interpreting the results because the error may be large in certain phases (e.g., the swing phase in this study).

Comment 4
In the introduction, Authors should mention some wearable sensors, such as Cell Reports Physical Science 2023, 4:101191, Nano Energy 2020, 67: 104228.

Authors’ response 4
Thank you for recommending the articles above and the constructive suggestion. As noted in our response during the first review stage, we have checked and red the articles you recommended. Although the studies are notable, it is difficult to cite them in the Introduction section because they did not evaluate kinematics.